# Maternal One-Carbon Metabolism during the Periconceptional Period and Human Foetal Brain Growth: A Systematic Review

**DOI:** 10.3390/genes12101634

**Published:** 2021-10-17

**Authors:** Eleonora Rubini, Inge M. M. Baijens, Alex Horánszky, Sam Schoenmakers, Kevin D. Sinclair, Melinda Zana, András Dinnyés, Régine P. M. Steegers-Theunissen, Melek Rousian

**Affiliations:** 1Department of Obstetrics and Gynecology, Erasmus MC, University Medical Center, 3000 CA Rotterdam, The Netherlands; e.rubini@erasmusmc.nl (E.R.); i.baijens@erasmusmc.nl (I.M.M.B.); s.schoenmakers@erasmusmc.nl (S.S.); m.rousian@erasmusmc.nl (M.R.); 2Department of Physiology and Animal Health, Institute of Physiology and Animal Health, Hungarian University of Agriculture and Life Sciences, H-2100 Gödöllő, Hungary; alex.horanszky@biotalentum.hu (A.H.); andras.dinnyes@biotalentum.hu (A.D.); 3BioTalentum Ltd., H-2100 Gödöllő, Hungary; melinda.zana@biotalentum.hu; 4School of Biosciences, University of Nottingham, Nottingham LE12 5RD, UK; kevin.sinclair@nottingham.ac.uk; 5HCEMM-USZ Stem Cell Research Group, Hungarian Centre of Excellence for Molecular Medicine, H-6723 Szeged, Hungary; 6Department of Cell Biology and Molecular Medicine, University of Szeged, H-6720 Szeged, Hungary

**Keywords:** one-carbon metabolism, folate, pregnancy, periconception, foetal programming, fetus, embryo development, brain, head

## Abstract

The maternal environment during the periconceptional period influences foetal growth and development, in part, via epigenetic mechanisms moderated by one-carbon metabolic pathways. During embryonic development, one-carbon metabolism is involved in brain development and neural programming. Derangements in one-carbon metabolism increase (i) the short-term risk of embryonic neural tube-related defects and (ii) long-term childhood behaviour, cognition, and autism spectrum disorders. Here we investigate the association between maternal one-carbon metabolism and foetal and neonatal brain growth and development. Database searching resulted in 26 articles eligible for inclusion. Maternal vitamin B_6_, vitamin B_12_, homocysteine, and choline were not associated with foetal and/or neonatal head growth. First-trimester maternal plasma folate within the normal range (>17 nmol/L) associated with increased foetal head size and head growth, and high erythrocyte folate (1538–1813 nmol/L) with increased cerebellar growth, whereas folate deficiency (<7 nmol/L) associated with a reduced foetal brain volume. Preconceptional folic acid supplement use and specific dietary patterns (associated with increased B vitamins and low homocysteine) increased foetal head size. Although early pregnancy maternal folate appears to be the most independent predictor of foetal brain growth, there is insufficient data to confirm the link between maternal folate and offspring risks for neurodevelopmental diseases.

## 1. Introduction

Life-long health is shaped during the early (embryonic) and late foetal stages of pregnancy, as proposed by the Developmental Origins of Health and Disease hypothesis [1]. Embryonic and foetal development are regulated by epigenetic mechanisms that are, in turn, dynamically modified by the status of the maternal and intrauterine environment [2,3]. Adverse conditions such as suboptimal maternal nutrition [4], gut dysbiosis [5], smoking [6], alcohol consumption [7], diabetes [8], and assisted reproductive technologies [9] can perturb embryonic/foetal epigenetic processes resulting in adverse long-term health outcomes, such as the increased risk of obesity [5], metabolic syndrome [10] and neurodevelopmental disorders [6,11] during postnatal life.

Substrates for epigenetic processes are supplied, in part, by one-carbon metabolism [12,13]. One-carbon metabolism is composed of the folate, methionine, and glutathione cycles (the latter not addressed in this review) and trans-sulphuration pathway, which provides one-carbon moieties indispensable for nucleotide and protein synthesis, and methylation of DNA and histones (Figure 1) [13]. Important nutrients that participate in one-carbon metabolism include choline (and betaine), methionine, vitamin B_6_, B_9/11_ (folate), and B_12_, all of which are supplied by diet or use of supplementation and are additionally synthetised by the gut microbiota [14].

The methylated status of DNA and associated histones strongly depends on the maternal supply of one-carbon metabolites. The epigenetic landscape is critical for proper lineage-specific differentiation and development of the foetal brain [15,16,17]. Perturbations in one-carbon metabolism during embryonic development can have a knock-on effect on the methylation signatures of neural cells in the developing brain, which in turn can lead to neurodevelopmental aberrations [18]. Severe maternal folate and vitamin B_12_ deficiency, and increased total homocysteine (tHcy) concentrations, during the periconceptional period, are associated with an increased risk of neural tube defects (NTDs) in the offspring [19,20,21]. However, in the absence of NTDs, subtle derangements in one-carbon metabolism can lead to long-term effects in the offspring, such as early childhood behaviour and learning deficits, and increased risk of psychiatric and autism spectrum disorders (ASD) presenting later in life [22,23,24,25]. It is known that predisposition to these deficits can occur prenatally during the period of brain programming, development, and growth. For example, abnormal foetal head growth has been proposed as an early biomarker for ASD, characterised by a reduced growth pattern prenatally followed by an accelerated pattern postnatally [26,27,28]. Currently, there is no consensus on the effects of maternal one-carbon metabolites on anatomical growth characteristics of prenatal brain development.

This article critically assesses our current understanding of the effects of maternal one-carbon metabolism during the periconceptional period and later stages of pregnancy on foetal and neonatal brain development and growth, with the aim of establishing a link between prenatal growth and postnatal risk of neurodevelopmental disorders. In the future, this may contribute to improving mental health and early prediction, treatment, and prevention of neurodevelopmental disorders in the offspring.

## 2. Materials and Methods

### 2.1. Search Strategy

A literature search was performed on Embase, Medline, Web of Science Core Collection, and Cochrane Central Register of Controlled Trials databases, including articles up to February 2021. The PRISMA protocol for systematic reviews was followed for article selection [29], and the review was registered to the PROSPERO registry (PROSPERO 2021 CRD42021239686). The search strategy included but was not limited to synonyms and closely related terms of the following Emtree terms: one-carbon metabolism, folic acid deficiency, nervous system development, brain growth, pregnancy, prenatal development, and growth (Appendix A).

### 2.2. Eligibility Criteria

Articles were eligible if they included measures of maternal one-carbon metabolites during the periconceptional period and/or pregnancy, and embryonic, foetal, and neonate head and brain measurements were reported. All human experimental studies, observational cohort studies, randomised controlled trials (RCTs), intervention studies, and case-control studies were eligible for inclusion, but only if written in the English language. Review articles and conference abstracts were excluded from the search. ER and IMMB reviewed the titles and abstracts, selected eligible full-text articles, and scored the articles independently.

### 2.3. Quality Assessment

The quality of eligible studies was assessed using the ErasmusAGE quality score system for systematic reviews [30]. Each article was assessed based on 5 items (study design, study size, exposure, outcome, and adjustments) and was given a score from zero to two per item to generate a total score from 0 to 10 per article, where 10 represented the highest quality score study (Appendix A).

## 3. Results

### 3.1. Study Selection

The literature search resulted in 951 unique records, of which 20 were added following manual database searching (Figure 2). Subsequently, 924 articles were excluded after title and abstract screening.

A total of 27 articles were reviewed for relevance, and only 1 was excluded because of missing data in the Results section. The 26 selected studies included cohort studies (*n* = 21), cross-sectional studies (*n* = 1) and RCTs (*n* = 4). One-carbon metabolism intermediates included in the selected articles were vitamin B_6_ (*n* = 1), B_9/11_ (folate, *n*= 20), B_12_ (*n* = 6), choline (*n* = 1), homocysteine (*n* = 6) and dietary patterns associated with one-carbon metabolism (*n* = 3). Article characteristics and quality scores are presented in Appendix A. According to the ErasmusAGE quality score system, the articles ranged from a score of 3 to a score of 8 (median 6). For a better interpretation of the results, articles were categorised into high- and low-quality scores according to the ErasmusAGE quality score system, whereby ≥6 was considered high quality and ≤5 of low quality. This categorisation was performed where possible according to the number of articles per maternal exposure factor.

### 3.2. Vitamin B_6_

The study by Takimoto et al. (ErasmusAGE score 4) described that neonatal HC was negatively correlated with maternal vitamin B_6_ during late pregnancy [31].

### 3.3. Vitamin B_9/11_ (Folate)

#### 3.3.1. Erythrocyte, Serum and Plasma Folate Concentrations

Ten studies examined the relationship between serum/plasma folate or red blood cell (RBC) folate concentrations and brain size, including one RCT [32], eight prospective cohort studies [33,34,35,36,37,38,39,40], and one cross-sectional study [41] (Table 1). Serum/plasma folate is indicative of recent folate intake, whereas RBC folate concentrations are useful to measure long-term folate status, as folate accumulates in RBCs during erythropoiesis and remains unvaried across the lifespan of the RBC (approximately 120 days) [42]. 

*High-quality score studies*. Steenweg-de-Graaff et al. showed no association between folate levels or folate deficiency and foetal HC at 20 weeks. However, a higher maternal plasma folate concentration (>17.4 nmol/L) was associated with larger foetal HC at 30 weeks of gestation but not at 20 weeks of gestation and at birth, and increased HC growth between 20 and 30 weeks of gestation. For each 1-standard deviation (SD; 1 SD folate = 8.8 nmol/L) of folate concentration, foetal head growth increased by 0.004 SD per week [34]. Bergen et al. reported no significant association between maternal folate and foetal HC in the second and third trimesters [40]. Zou et al. found that brain volumes of foetuses from mothers with folate deficiency during early pregnancy were smaller in the third trimester when compared to brain volumes of foetuses from mothers without folate deficiency [35]. Nilsen et al. reported no significant associations with neonatal HC [37]. Koning et al. reported that foetal cerebellar growth in the first trimester was significantly highest in the third quartile of RBC folate concentrations (1538–1813 nmol/L, control) compared to higher or lower concentrations [36].

*Low-quality score studies*. In two articles, a positive association between RBC folate levels in early pregnancy and neonatal HC was described [32,33]. Similar associations were found with serum folate in one study (*R* = 0.394, *P* = 0.044) [39]. This was in contrast to two other studies, which found no association between serum, plasma, and RBC folate concentrations and neonatal HC across all trimesters [38,41]. RBC folate was positively associated with neonatal HC during the second trimester only [38].

*Conclusion*. Maternal plasma folate concentrations above 17 nmol/L are associated with larger foetal head size and growth from mid-to-late pregnancy, and RBC folate within the range 1538–1813 nmol/L is associated with increased foetal cerebellar size in the first trimester. In contrast, folate deficiency is associated with a smaller brain volume during pregnancy. There is no evidence of associations between maternal blood folate and neonatal head size at birth.

#### 3.3.2. Dietary Intake of Folate

Three studies investigated maternal dietary intake of folate via questionnaires and neonatal HC (Table 2).

*High-quality score studies.* According to the study by Nilsen et al., daily intake of folate throughout pregnancy was not associated with neonatal HC [37].

*Low-quality score studies.* Two studies agreed that prenatal daily intake of folate, specifically during early or late pregnancy, was not associated with neonatal HC [33,38].

*Conclusion*. There is no association between maternal dietary intake of folate and neonatal HC.

#### 3.3.3. Folic Acid Supplement Use

Fourteen studies investigated the effects of maternal folic acid supplement use and the time point at which supplementation was initiated with respect to conception on foetal and neonatal brain growth and size (Table 2).

*High-quality score studies*. Four articles studied prenatal foetal head growth. Steenweg-de-Graaf et al. showed that foetuses of mothers who commenced folic acid supplement use preconceptionally had a significantly slightly larger head size and circumference at 20 weeks of gestation, but there was no evidence of an association between maternal folic acid supplement use during pregnancy and prenatal head growth [34]. Periconceptional folic acid use, compared to women who did not use folic acid, was not significantly associated with foetal HC in mid and late pregnancy [43]. Koning et al. reported significantly increased proportional growth for TCD, RCD, and LCD diameters in the first trimester of pregnancy when maternal folic acid supplement use was initiated preconceptionally instead of postconceptionally [36]. Regarding other brain structures, Husen et al. demonstrated that preconceptional initiation of folic acid supplement use was not associated with foetal diencephalon total diameter, mesencephalon total diameter, left telencephalon thickness, or right telencephalon thickness measurements at either 9 or 11 weeks [48].

Five articles studied neonatal HC. Preconception, early and mid-trimester folic acid supplement use were not associated with neonatal HC in two studies [37,46]. An RCT by Yusuf et al. reported that even high dose folic acid use (4 mg/day) during the first trimester among smokers had no significant effect on neonatal HC or brain weight at birth when compared to those who received the standard dosage (0.8 mg/day) [44]. However, folic acid supplement use in combination with iron from preconception onwards significantly increased HC at birth by 0.16 cm (*B* = 0.16, 95% CI −0.03; 0.34, *P* = 0.012) [45]. An alternative supplement to folic acid is 5-methyltetrahydrofolate, which is a downstream product within the folate cycle. 5-methyltetrahydrofolate supplement use at 22 weeks of gestation had no association with neonatal HC at birth [47].

*Low-quality score studies*. The studies of Takimoto et al. and Hossein-nezad et al. showed that maternal folate intake across all trimesters, and additionally the time point at which supplementation was initiated with respect to conception, was not associated with neonatal HC [38,51]. An exception was the study of Takimoto et al. [31]. Female neonatal HC at birth in that study was lower among folic acid supplement users during the first and second trimester than in non-users. One study revealed a significant increase in foetal mesencephalon-to-occiput distance in the first trimester in women who did not receive folic acid supplementation [50]. Koning et al. reported no significant association between growth trajectories of the foetal cerebellum during the first trimester of pregnancy and the time point at which supplementation was initiated with respect to conception [49].

*Conclusion**s*. Preconceptional, as opposed to periconceptional, folic acid supplement use is associated with larger head size by mid-pregnancy and larger cerebellar size during early pregnancy, but not with larger brain size in early pregnancy nor with head size at birth.

### 3.4. Vitamin B_12_

Six studies investigated the association between maternal vitamin B_12_ during pregnancy and foetal or neonatal head measurements, five analysed vitamin B_12_ from serum/plasma, and two from dietary intake (Table 3).

*High-quality score studies*. Bergen et al. reported no association between vitamin B_12_ in early pregnancy and neonatal head circumference (HC) [40]. Similarly, Tan et al. revealed that first- and second-trimester maternal total vitamin B_12_ was not linearly associated with neonatal HC Z-score [53].

*Low-quality score studies*. According to two studies, maternal plasma and serum vitamin B_12_ concentrations across all three trimesters and in the third trimester only were not associated with neonatal HC at birth [38,41]. Only the study by Jiang et al. reported that maternal serum vitamin B_12_ levels were positively correlated with neonatal HC at birth, although the timing of blood analysis was not defined [39]. Two studies analysed dietary sources of vitamin B_12_ across all three trimesters of pregnancy, together with neonatal HC measurements, but also observed no significant associations [38,54].

*Conclusion*. No association between maternal vitamin B_12_ and neonatal head size.

### 3.5. Choline

The study by Nakanishi et al. (ErasmusAGE score 5) revealed that higher maternal plasma choline levels at term were significantly associated with reduced neonatal HC [55].

### 3.6. Total Homocysteine

Six studies reported associations between maternal tHcy concentrations during pregnancy and neonatal head size (Table 4).

*High-quality score studies.* Bergen et al. reported that maternal plasma tHcy concentrations during early pregnancy were not associated with foetal HC from late pregnancy to birth [40]. In the study by Nilsen et al., no significant association with neonatal HC was observed when tHcy concentrations were analysed during mid-pregnancy [37]. Similarly, Tan et al. reported that maternal tHcy concentrations during either the first or second trimester were not linearly associated with neonatal HC Z-score [53].

*Low-quality score studies*. Two studies reported that second- and third-trimester pregnancy tHcy concentrations were positively correlated with neonatal HC [31,38]. In one study, maternal serum tHcy levels were found to be negatively correlated with neonatal HC, although the timing of assessment was not reported [39].

*Conclusion.* No association between maternal tHcy and foetal head size from mid-pregnancy to birth.

### 3.7. Maternal Dietary Patterns

Three high-quality score articles found associations between maternal dietary patterns during early and mid-pregnancy and one-carbon metabolites and analysed their influence on foetal brain and head measurements (Table 5). Parisi et al. found that strong adherence to a dairy-rich dietary pattern was associated with significantly lower plasma tHcy concentrations during early pregnancy compared to weak adherence [56]. In addition, vitamin B_2_, B_6_, B_12,_ and folate nutrient intakes were significantly correlated to a dairy-rich dietary pattern. Maternal adherence to a dairy-rich dietary pattern was also associated with increased foetal TCD measurements from first to third trimester. In the article by Timmermans et al., higher maternal tHcy concentrations (median 7.1 µmol/L), lower serum vitamin B_12_ (median 168.5 pmol/L), and folate concentrations (median 18.2 nmol/L) during early pregnancy were observed among women with low adherence to a Mediterranean diet, which had a smaller foetal HC in late pregnancy [57]. Lastly, Lecorguillé et al. concluded that a ‘varied and balanced’ dietary pattern (defined by high intake of dairy, proteins, and vegetables and low intake of snacks and sweetened beverages) was positively associated with dietary B vitamins, choline, and methionine. Adherence to this diet during the second trimester of pregnancy, however, was not significantly associated with neonatal HC at birth [52].

*Conclusion.* Strong adherence to a dairy-rich dietary pattern and low adherence to a Mediterranean dietary pattern is associated, respectively, with low and high levels of tHcy, high and low levels of B vitamins, and with an increased foetal cerebellum and head size during pregnancy.

## 4. Discussion

This review systematically summarises associations between maternal one-carbon metabolism during the periconceptional period and pregnancy and foetal and neonatal brain growth. Studies including one-carbon metabolism intermediates, which were assessed in blood or derived from diet, were included when they related these determinants to the outcome foetal and/or neonatal brain structure sizes, HC, and growth patterns of these measures. One-carbon metabolism is a complex pathway of biochemical reactions that involve the transfer of 1 carbon unit to provide substrates for cellular processes [58]. Most of these substrates are crucial for brain development and function, such as the biosynthesis of biogenic amines, phospholipids (particularly long-chain polyunsaturated fatty acids), and creatine, which are fundamental for neurotransmitter activity and structural neural integrity [58]. Additionally, the folate and methionine cycles are necessary for genomic maintenance of brain cell populations and epigenetic control of gene expression via the production of methyl groups for DNA and histone methylation [58]. Animal studies demonstrate that one-carbon derangements, particularly folate deficiency, are associated with expression changes of DNA methyltransferases and reduction in cell proliferation in the brain [59]. In humans, faulty regulation of these epigenetic processes is associated with the onset of several neurodevelopmental disorders [18]. However, the molecular mechanisms that explain the role of one-carbon metabolism, brain development, and long-term brain function are currently undefined.

Derangements in maternal one-carbon metabolism are associated with childhood behaviour and cognition problems and to psychosis and ASD later in life [22]. Atypical foetal brain growth patterns are considered an early biomarker for learning or behaviour difficulties, as children with ASD and language impairment exhibit reduced HC growth patterns prenatally followed by catch-up growth postnatally [60,61,62,63]. Currently, the most common measurement of foetal brain growth is HC at birth, which is acknowledged as a non-invasive marker for brain volume and growth in a clinical setting [64]. However, with the introduction of more advanced medical technologies, prenatal imaging of head and brain structures is being introduced into clinical research, providing better insights into intrauterine foetal development [65]. In this article, we assessed our current understanding of the effects of maternal one-carbon metabolism during the periconceptional period and later stages of pregnancy on foetal and neonatal brain development and growth, with the aim of establishing a link between prenatal growth and postnatal risk of neurodevelopmental disorders. Interpretation of the articles was based on the ErasmusAGE quality score ranking.

### 4.1. One-Carbon Metabolism

In this article, we found that maternal folate, folic acid supplement use, and dietary patterns (associated with the one-carbon metabolism) played a role in foetal and/or neonatal head growth, but this did not apply to maternal vitamin B_6_, vitamin B_12_, homocysteine and choline (Figure 3). Interpretation of the results was based on general reference values of one-carbon metabolism intermediates determined from blood (Table 6).

#### 4.1.1. Folate

The role of folate and DNA methylation has been extensively investigated. Several folate coenzymes are critical for methyl group delivery [70]. In this review, we found three out of four high-quality score articles supporting an association between maternal folate concentrations and foetal head growth. Maternal plasma folate concentrations above >17 nmol/L, which lie within the normal range values (Table 6), were associated with larger foetal head size and growth from mid to late pregnancy. Instead, high maternal RBC folate in the ranges of 1538–1813 nmol/L, which lie above the recommended value for NTD prevention (Table 6), was associated with increased foetal cerebellar size in the first trimester. The foetal cerebellum is a rapidly developing brain structure, involved in cognitive and language behaviour postnatally and associated with postnatal neurodevelopment and mental health [71]. There is no clear evidence of any side-effects caused by excessive RBC folate during pregnancy; however, knowing that folate has critical roles in shaping the offspring epigenome, as evident in the well-known experiment on Agouti mouse coat colour [72], we must be conscious that exceeding recommended values may have underlying biological consequences still unknown. In agreement with such, folate deficiency (<7 nmol/L) was associated with a smaller brain volume during pregnancy [34,35,36]. With the exception of low-quality studies, there was no evidence of associations between maternal blood folate and neonatal head size at birth. Maternal dietary folate was not associated with neonatal head size [37].

From infancy to late childhood, there is an emerging consensus in the literature that higher maternal folate concentrations improve offspring language performance scores, cognitive scores and increase brain volume [23,35,73,74]. However, a few studies report no associations [75,76]. Overall, we speculate that plasma and RBC folate during early pregnancy may be the most significant biomarkers to predict foetal brain growth. Instead, the role of maternal folate and the neonatal brain remains undefined. What is reported in this review and in the literature supports the hypothesis that early pregnancy folate concentrations may promote prenatal brain development and so, consequently, may improve language and cognitive scores in childhood. However, the literature is inconsistent, so further research is required to understand the role of folate on short- and long-term offspring neurological development and health.

#### 4.1.2. Folic Acid Supplement Use

It is known from animal studies that folic acid supplementation impacts DNA methylation implicated in neurodevelopment and learning/memory abilities [77]. In this review, evidence of the benefit of folic acid supplement use, and the timing at which supplementation was initiated, on foetal brain development was inconsistent. Preconception folic acid supplement use increased foetal head size and cerebellar growth in three studies [34,36,45], but not in two other studies [46,48]. Folic acid supplement use at other time points had no effect on neonatal head size [37,43,44,47]. The majority of studies that assessed maternal folic acid supplementation disclosed that supplementation and food fortification, even after conception, ameliorated offspring neurodevelopment and reduced the risk of psychosis from infancy to childhood [78,79,80,81,82]. In epidemiological studies, determining the role of folic acid supplement use is challenging, particularly as there is often limited knowledge on the nutritional status of the patient, particularly if there is unknown use of multivitamin supplements. Despite insufficient evidence, we can only hypothesise that preconceptional folic acid supplement use, as opposed to periconceptional use, is associated with prenatal brain development. However, its use at any stage of pregnancy is sufficient to reduce the risk of long-term poor cognitive and behaviour performance in offspring.

#### 4.1.3. Dietary Patterns

From the results obtained, two out of three articles state that a diet associated with low tHcy and high vitamin B concentrations, commencing during early pregnancy, is associated with increased foetal head and cerebellar size [56,57]. Mothers who consumed diets rich in fruits and vegetables had offspring with higher average IQ scores, compared to other dietary patterns, and with lower risks of exhibiting negative neurobehavioral effects [83,84]. This is observed in Mediterranean dietary patterns, rich in fruits and vegetables, which are a suitable source of B vitamins. In agreement with our results, a vitamin B-rich dietary pattern such as the Mediterranean diet, from early pregnancy onwards, may favour prenatal and postnatal offspring neurodevelopment.

#### 4.1.4. Vitamin B_6_

According to this review, there is insufficient data to determine the role of maternal vitamin B_6_ on foetal neurodevelopment. In addition, the literature investigating the role of maternal vitamin B_6_ and infant long-term neurological health remains scarce. Further research is required to address the role of maternal vitamin B_6_ and offspring neurological health.

#### 4.1.5. Vitamin B_12_

We found that maternal vitamin B_12_ was not associated with neonatal HC, regardless of its source and timing of assessment [40,53]. This can only be speculated as not all included articles defined vitamin B_12_ concentrations or divided them into concentration ranges covering normal reference values (Table 6). This is in agreement with the studies of Golding et al. and Srinivasan et al., which report no significant association between maternal vitamin B_12_ and vocabulary and language scores at 1–2 years of age, and no difference in infant Bayley Scales of Infant and Toddler Development (BSID) scales upon maternal vitamin B_12_ supplementation, respectively [85,86]. In contrast, two studies report a positive correlation between vitamin B_12_ and cognition-based scores, specifically at 1–2 years of age [87,88]. Regarding long-term effects, no association of maternal vitamin B_12_ and child cognitive performance and IQ is observed [74,89]; however, lower attention and decreased short-term memory scores were found in 9-year-old children in the study by Bhate et al. [90]. Only extremely high maternal vitamin B_12_ levels were associated with ASD risk [24]. Overall, there is no clear agreement on the role of maternal vitamin B_12_ on offspring neurodevelopment, from prenatal to postnatal development.

#### 4.1.6. Homocysteine

The methylation of homocysteine to methionine precedes the generation of S-adenosylmethionine, which is a critical methyl donor for many functional reactions and in the development of the brain [91]. Homocysteine methylation is dependent amongst other substrates and cofactors upon vitamin B_12_, folate, betaine, and choline, and perturbations in these substrates/cofactors can result in disturbances in methionine metabolism and altered methylation accompanied by increases of tHcy. It is well documented that elevated tHcy during adulthood is associated with cognitive impairments [92], but much less is known of the effects in early and late foetal life. In this review, maternal tHcy within the normal range (Table 6) was not associated with foetal or neonatal HC, regardless of the timing of assessment [37,40,53]. However, studies agree that elevated tHcy levels, assessed preconceptionally and during pregnancy, are negatively associated with low scores of the BSID test in the first year of life, particularly expressive language and fine motor behaviour [85,86]. Regarding long-term effects, there appears to be no association of maternal tHcy with childhood cognitive performance [74,75]. Only Ars et al. describe that prenatal maternal high tHcy levels predicted poorer performance in language and visuo-spatial tests of children aged 6–8 years [23]. Overall, there is no agreement on the role of maternal tHcy on offspring neurodevelopment based on phenotypical measurements. This does not exclude that the effects of high maternal tHcy may have an effect at the molecular (epigenetic) level and manifest phenotypically only in postnatal life.

#### 4.1.7. Choline

Studies show that maternal choline levels modulate DNA and histone methylation in offspring, which suggests that there are long-term implications for neurodevelopmental processes induced as a result of fluctuations in maternal choline supply [93]. As found in this review, there is no consensus on the effects of maternal choline on foetal or child neurodevelopment. One study reports no effect on child IQ or on any memory or visuospatial tests [94], whereas another study reports that high second-trimester choline concentrations were associated with improved child memory and learning scores [76]. The roles of maternal choline during pregnancy and foetal neurodevelopment remain undefined and require further investigation.

### 4.2. Main Interpretation

This review highlights the key role of the maternal one-carbon metabolism during the periconceptional period, and particularly blood folate, in the programming of foetal brain development. Suboptimal periconceptional maternal blood folate levels, which are determined amongst others by dietary, lifestyle, metabolic and genetic factors, may influence epigenetic programming of foetal brain development, disrupt brain growth trajectories, and increase the risk of long-term neurological deficits in the offspring (Figure 3). We have noticed that maternal serum folate concentrations within the normal range, and high RBC folate concentrations in early pregnancy, may accelerate foetal brain growth and increase brain size up to late pregnancy. This likely represents a positive effect, as increased brain growth may minimise the risks of atypical head growth trajectories common in children with ASD-related deficits. However, it remains challenging to interpret the role of accelerated prenatal brain growth, and it is currently undefined whether it is a representation of a beneficial or undesirable effect. Blood folate levels can be increased by dietary patterns such as the Mediterranean and dairy-rich diets and by preconceptional folic acid supplement use, which have been reported as beneficial in supporting increased foetal head size during pregnancy. This was not the case for other one-carbon metabolism intermediates.

### 4.3. Clinical Relevance and Future Research

Women are advised to follow a healthy diet and take pregnancy supplements as early as possible when planning a pregnancy. However, this does not guarantee protection from derangements in one-carbon metabolism at the time of pregnancy. Genetic factors, individual metabolism, and lifestyle habits are a number of additional factors that contribute to the way B vitamins are stored and circulate in the body. As reported in this review, even subtle derangements may be harmful to offspring. Extra care should be implemented for pregnant or pregnant-to-be women, such as routine blood tests for concentrations of one-carbon metabolites when planning a pregnancy or, at the latest, once the pregnancy is announced.

From this review, we can state that there is limited information on the relationship between prenatal brain growth determinants and trajectories and postnatal neurological development assessed by behaviour tests. It is recommended that more research is undertaken to investigate associations between early biomarkers during early and late foetal development and long-term health. Anticipated diagnosis as early as the preconceptional stages of development would be a powerful tool to anticipate pregnancy complications or plan postnatal treatment and adequate follow-up. The current state of new medical technologies allows for detailed prenatal foetal growth assessment performed in a non-invasive way. Prenatal screening of the foetal brain and its growth trajectory, with interpretation for postnatal outcomes, would be valuable for clinical care. Meanwhile, enhancing our understanding of foetal epigenetic alterations induced by disturbed maternal one-carbon metabolism during brain development would benefit the application of preventative or follow-up measures or therapies to reduce the risk of neurodevelopmental disorders.

### 4.4. Strengths and Limitations

The present article is the first to systematically review our current state of knowledge on the impact of maternal one-carbon metabolism during periconception and pregnancy and foetal and neonatal brain and head development. The available literature mostly focuses on the postnatal effects of derangements in maternal one-carbon metabolism. Information on prenatal development is currently sparse. For this reason, only foetal brain or head measurements reported during pregnancy until birth were included in these analyses. Despite an extensive literature search, the number of included articles and those of high-quality scores was low. In some cases, small sample sizes and non-reported information were major limitations to the quality of the study. The diversity among exposure and outcome measurement time points limited the possibility of making equivalent comparisons between studies, thus reducing the strength of the observations. Similarly, the restricted selection of foetal brain structures analysed among the studies did not allow a comprehensive overview of foetal brain development, which remains to be addressed in future studies. However, comparison among high-quality score articles was possible as the majority of them included B vitamin concentrations within normal reference values. This did not apply to low-quality score articles as not all articles stated the exact biomarker concentrations to be able to draw a conclusion on the effect of concentration ranges and outcome variables. The roles of other intermediates may have been masked by the cyclical nature of the pathway, redundancy, and their instability and should be addressed in future research. We also cannot exclude that the results stated may derive from a synergistic or summative effect of multiple micronutrients not included in the analysed variables. Lastly, as this review includes only gross phenotypic measurements of the brain or head structures, we cannot exclude the possibility of more subtle microstructural or cellular/molecular alterations. Nevertheless, this review provides evidence that subtle alterations to one-carbon metabolism can influence foetal brain growth.

## 5. Conclusions

Folate is a suitable marker for the one-carbon metabolism status. Maternal plasma and RBC folate concentrations in early pregnancy may be a marker for postnatal neurologic physiology and pathophysiology. Normal range plasma folate and high RBC folate concentrations (plasma folate >17 nmol/L and RBC folate 1538–1813 nmol/L) may accelerate foetal head growth and increase size, thus possibly minimising the risk of neurological diseases postnatally. However, there is limited information on prenatal foetal brain development in response to maternal one-carbon metabolism and long-term offspring health. In clinical care, women contemplating pregnancy or that are of reproductive age should be tested for blood folate concentrations and diets adjusted or appropriately supplemented in order to ensure that blood folate levels are within the normal range.

## Figures and Tables

**Figure 1 genes-12-01634-f001:**
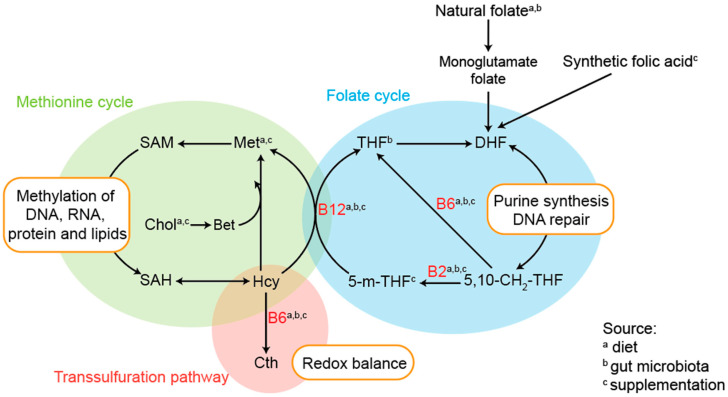
One-carbon metabolism (depicting the three core pathways) and main metabolic/molecular outputs. Substrates: betaine (Bet), choline (Chol), cystathione (Cth), dihydrofolate (DHL), homocysteine (Hcy), 5,10-methenyl-tetrahydrofolate (5,10-CH2-THF), 5-methyl-tetrahydrofolate (5-m-THF), methionine (Met), tetrahydrofolate (THF), S-adenosylmethionine (SAM), S-adenosylhomocysteine (SAH). Cofactors: vitamin B_2_ (B2), vitamin B_6_ (B6), vitamin B_12_ (B12). Outputs: substrate methylation, redox balance, purine synthesis, and DNA repair.

**Figure 2 genes-12-01634-f002:**
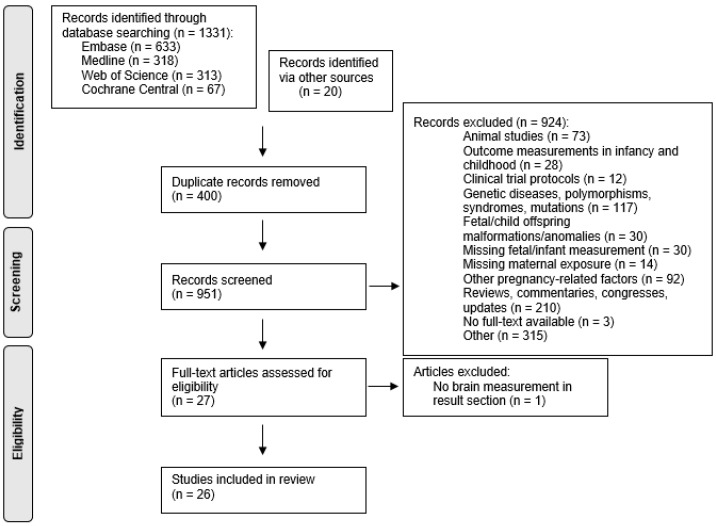
Flowchart of included and excluded articles following the PRISMA guidelines.

**Figure 3 genes-12-01634-f003:**
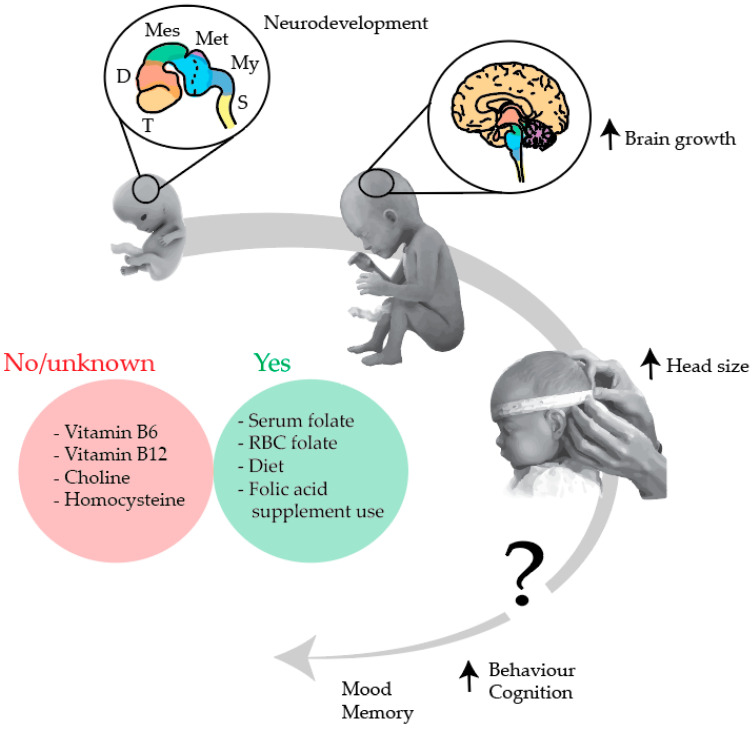
Maternal one-carbon metabolism and the associations with prenatal brain development and neonatal head growth. In this review, only maternal serum folate, high RBC folate, diet, and folic acid supplementation (Yes), as opposed to vitamin B_6_, B_12_, choline, and homocysteine (No/unknown), were shown to accelerate prenatal brain growth and increase neonatal head size. The link between prenatal growth and postnatal behaviour is currently undefined. The figure represents a timeline starting from neurodevelopment patterning in the first trimester to brain growth in late pregnancy, head size at birth, and behaviour, cognition, mood, and memory in childhood and adulthood. Brain structures are colour-coded: during pregnancy, the telencephalon develops into the cerebrum, the diencephalon into the thalamus, the mesencephalon into the midbrain, the metencephalon into the pons and cerebellum, and the myelencephalon into the medulla. Abbreviations: diencephalon (D), mesencephalon (Mes), metencephalon (Met), myelencephalon (My), red blood cell (RBC), spinal cord (S), telencephalon (T). Image adapted from A.D.A.M. Inc. [69].

**Table 1 genes-12-01634-t001:** Description and summary of 10 studies that investigated associations between maternal folate concentrations during pregnancy and early and late foetal or neonatal head measurements ordered according to the quality score.

Author (Year)	Exposure (Range)	Exposure Timing	Outcome(s)	Association	Statistical Value	Quality Score
Bergen et al. (2016) [40]	Plasma folate (6.2–34.3 nmol/L)	Early pregnancy (median 13.5 weeks GA)	Trend for folate in low quintiles (<13.10 nnmol/L) associated with reduced foetal HC in 2nd and 3rd trimester and birth	=	Q1 (≤9.10 nmol/L) *B* = −2.0 Q2 (9.11–13.10) *B* = −1.2 Q3 (13.11–18.90) *B* = −0.2 Q4 (18.91–25.80) *B* = −1.3 *P* = 0.14	8
Steenweg-de Graaff et al. (2017) [34]	Plasma folate (1.8–45.3 nmol/L)	Early pregnancy (13.2 weeks GA)	Higher folate associated with larger foetal HC 30 weeks GA but not at 20 weeks and at birthHigher folate associated with increased foetal head growth between 2nd and 3rd trimester	+	*B* = 0.47, *P* ≤ 0.001 ^f^ *B* = 0.004, *P* = 0.02 ^g^	8
Nilsen et al. (2010) [37]	Plasma folate (<5.9->14.8 nmol/L)	Mid-pregnancy (median 18 weeks GA)	No linear association with neonatal HC at birth	=	*P* = 0.53	7
Zou et al. (2020) [35]	Plasma folate (1.8–45.3 nmol/L)	Early pregnancy (mean 13.3 weeks GA)	Folate deficiency associated with reduced foetal brain volume from 3rd trimester to childhood	−	*B* = −0.04, *P* = 0.02	7
Koning et al. (2015) [36]	RBC folate (814–2936 nmol/L)	1st trimester (≤ 8+0 weeks GA)	Highest foetal proportional cerebellar growth and TCD, RCD, and LCD was highest in the third quartile of RBC folate (1538–1813 nmol/L) in 1st trimester	+	Q1 (−0.0721 mm/day) *P* < 0.01 ^c^ Q2 (−0.0438) *P* < 0.01 ^c^ Q4 (−0.0459) *P* = 0.05 ^c^ Q1 (−0.0364 mm/mm) *P* < 0.05 ^d^ Q2 (−0.0280) *P* > 0.05 ^d^ Q4 (−0.0232) *P* > 0.05 ^d^ Q1 (−0.0017 mm/day) *P* >0.05 ^e^ Q2 (−0.0010) *P* > 0.05 ^e^ Q4 (−0.0459) *P* < 0.05 ^e^	6
Brough et al. (2010) [32]	RBC folate (Not reported)	Early pregnancy (>13 weeks GA)	Positively correlated with neonatal HC at birth	+	*R* = 0.111, *P* = 0.046	5
Schlotz et al. (2010) [33]	RBC folate (373.5–588.5 µg/L)	Median 95 days GA	Association with neonatal HC at birth	+	*Β* = 0.17, *P* = 0.031	5
Gadgil et al. (2014) [41]	Plasma folate (15.1–18.95 ng/mL)	Late pregnancy (36 weeks GA)	No correlation with neonatal HC at birth Higher folate-to-vitamin B_12_ ratio correlated with decreased neonatal HC at birth	= (- folate-to-vitamin B_12_)	*R* = −0.089, *P* = 0.672 ^a^ *R* = −0.469, *P* = 0.018 ^b^	4
Jiang et al. (2016) [39]	Serum folate (Not reported)	Not reported	Positively correlated with neonatal HC at birth	+	*R* = 0.394, *P* = 0.044	3
Takimoto et al. (2007) [38]	Serum and RBC folate (Not reported)	1st, 2nd, and 3rd trimester	No associations with neonatal HC at birth apart from RBC folate in the 2nd trimester	= (+ RBC folate 2nd trimester)	Serum folate: *ES* = 0.08 *P* = 0.09 ^h^ *ES* = −0.03 *P* = 0.07 ^i^ *ES* = 0.004 *P* = 0.57 ^j^ RBC folate: *ES* = −0.0003 *P* = 0.56 ^h^ *ES* = 0.0005 *P* = 0.01 ^i^ *ES* = −0.0003 *P* = 0.90 ^j^	3

Abbreviations: effect estimates (ES), gestational age (GA), head circumference (HC), red blood cell (RBC), right cerebellar diameter (RCD), left cerebellar diameter (LCD), transcerebellar diameter (TCD), quartile (Q). Significant associations between the exposure and the outcome are reported as + if positive and − if negative. = indicates no significant association. ^a^ folate, ^b^ folate-to- vitamin B_12_ ratio, ^c^ as a function of GA, ^d^ as a function of CRL, ^e^ proportional growth, ^f^ head size, ^g^ head growth, ^h^ first trimester, ^i^ second trimester, ^j^ third trimester.

**Table 2 genes-12-01634-t002:** Description and summary of 17 studies that investigated associations between maternal dietary folate intake and folic acid supplement use during pregnancy and foetal or neonatal head measurements ordered according to the quality score.

Author (Year)	Exposure	Exposure Timing	Outcome(s)	Association	Statistical Value	Quality Score
Dietary intake						
Nilsen et al. (2010) [37]	Daily mean intake of dietary folate from FFQs	Throughout pregnancy	No association with neonatal HC at birth	=	*P* = 0.27	7
Schlotz et al. (2010) [33]	Dietary folate intake based on FFQs	Early (median gestational day 101) and late pregnancy (median gestational day 199)	Trend for association with neonatal HC at birth	=	*B* = 0.15, *P* = 0.083	5
Takimoto et al. (2007) [38]	Daily intake of dietary folate calculated from the Standard Food Consumption Table	Prenatal	No relation with neonatal HC at birth	=	Not reported	3
Supplement use						
Steenweg-de-Graaff et al. (2017) [34]	Folic acid supplement use via self-administered questionnaires	Early pregnancy (median 13.2 weeks GA)	Preconceptional supplement use slightly increased foetal head size and circumference at 20 weeks of gestation No association with foetal head growth	+ (= foetal head growth)	*B* = 0.112, *P* = 0.01 ^g^ *B* = 0.120, *P* = 0.01 ^h^	8
Timmermans et al. (2009) [43]	Folic acid supplement use via self-administered questionnaires	Mid-pregnancy (median 15.4 weeks GA)	Periconceptional supplement use was associated with trends towards larger foetal HC at 20 and 30 weeks of pregnancy	+	*B* = 0.61, *P* > 0.05 ^i^ *B* = 1.34, *P* > 0.05 ^j^	8
Yusuf et al. (2019) [44]	0.8 mg (control) or 4 mg (high dose, intervention) folic acid supplement per day	1st trimester (mean 12.3 weeks GA)	Higher dose users had a 1.88 mm larger neonatal HC at birth Higher dose had no effect on neonatal brain weight at birth	+ (= brain weight at birth)	*P* = 0.28 ^h^ 6.90 ± 5.85 g, *P* = 0.24 ^k^	8
Christian et al. (2003) [45]	Folic acid (400 µg/day), folic acid-iron (60 mg ferrous fumarate), folic acid-iron-zinc (30 mg zinc sulphate)	Preconception until birth	Folic acid-iron supplement use increased neonatal HC at birth by 0.16 cm	+	*B* = 0.16, *P* = 0.012	7
Nilsen et al. (2010) [37]	Folic acid supplement use via self-administered questionnaires	From start to mid-pregnancy (median 18 weeks GA)	No association with neonatal HC at birth	=	*P* = 0.44	7
Bulloch et al. (2020) [46]	Folic acid supplement use via lifestyle questionnaires	Preconception and at 15 + 1 weeks GA	No association with neonatal HC z-score at birth	=	*B* = 0.04, *P* = 0.197 ^a^ *B* = 0.01, *P* = 0.662 ^b^	6
Catena et al. (2019) [47]	5-m-THF supplement use	From 22 weeks GA to delivery	No effect on neonatal HC at birth		*P* > 0.13	6
Husen et al. (2021) [48]	Folic acid supplement use via self-reported questionnaires	Early pregnancy (<10 weeks GA)	Preconceptional initiation was not associated with either 9 or 11 weeks foetal DTD, MTD, TTL, or TTR measurements	=	DTD: *B* = 0.093, *P* = 0.422 ^c^ *B* = −0.068, *P* = 0.609 ^d^ MTD: *B* = −0.025, *P* = 0.814 ^c^ *B* = −0.004, *P* = 0.968 ^d^ TTL: *B* = 0.047, *P* = 0.141 ^c^ *B* = 0.027, *P* = 0.570 ^d^ TTR: *B* = 0.028, *P* = 0.531 ^c^ *B* = 0.025, *P* = 0.599 ^d^	6
Koning et al. (2015) [36]	Folic acid supplement use via self-administered questionnaires	1st trimester (≤8 + 0 weeks GA)	Preconceptional supplement use increased proportional foetal cerebellar growth for TCD, RCD, and LCD in 1st trimester	+	TCD: *B* = 0.257, *P* = 0.032 ^e^ *B* = −0.078, *P* = 0.616 ^f^ RCD: *B* = 0.156, *P* = 0.015 ^e^ *B* = 0.008, *P* = 0.918 ^f^ LCD: *B* = 0.171, *P* = 0.013 ^e^ *B* = 0.041, *P* = 0.601 ^f^	6
Koning et al. (2017) [49]	Folic acid supplement use via self-administered questionnaires	1st trimester (≤12 weeks GA)	No associations with foetal cerebellum growth trajectories in the 1st trimester	=	*B* = 0.22, *P* = 0.19	5
Nemescu et al. (2020) [50]	Folic acid supplement use	1st trimester (11–13 weeks GA)	No supplement use increased foetal MO in the 1st trimester	−	*P* = 0.014	5
Hossein-nezhad et al. (2011) [51]	Group 1: 1 mg/day folic acid in 1st and 2nd trimesters and Group 2: 1 mg/day folic acid until birth	Throughout pregnancy	No effect of timing of supplement use on neonatal HC at birth	=	*P* = 0.5	4
Takimoto et al. (2011) [31]	Folic acid supplement use via 24 h dietary recall survey	Throughout pregnancy	Decreased HC at birth in female neonates	−	*B* = 0.112, *P* = 0.01	4
Takimoto et al. (2007) [38]	Folic acid use from self-administered questionnaires	1st, 2nd, and 3rd trimesters	Not related to neonatal HC at birth	=	Not reported	3

Abbreviations: biparietal diameter (BPD), crown-to-rump length (CRL), diencephalon total diameter (DTD), food frequency questionnaire (FFQ), gestational age (GA), head circumference (HC), mesencephalon-to-occiput distance (MO), mesencephalon total diameter (MTD), left telencephalon thickness (TTL), right telencephalon thickness (TTR). Significant associations between the exposure and the outcome are reported as + if positive and − if negative. = indicates no significant association. ^a^ preconception use, ^b^ 15 weeks, ^c^ 9 weeks of gestation, ^d^ 11 weeks of gestation, ^e^ per mm increase in CRL, ^f^ per days increase in GA, ^g^ head size, ^h^ head circumference, ^i^ 20 weeks, ^j^ 30 weeks, ^k^ brain weight.

**Table 3 genes-12-01634-t003:** Description and summary of 6 studies that investigated associations between maternal vitamin B_12_ during pregnancy and neonatal head measurements ordered according to the quality score.

Author (Year)	Exposure (Range)	Exposure Timing	Outcome(s)	Association	Statistical Value	Quality Score
Serum/plasma						
Bergen et al. (2016) [40]	Non-fasting serum total and active vitamin B_12_ (83–315 and 20–83 pmol/L)	Early pregnancy (median 13.5 GA)	No associations with neonatal HC at birth	=	Q1 (≤119.0 pmol/L) *B* = −0.6 Q2 (119.0–153.0) *B* = −0.4 Q3 (153.01–189.0) *B* = 0.5 Q4 (189.01–244.0) *B* = −0.3, *P* = 0.28 ^a^ Q1 (≤30 pmol/L) *B* = −0.7 Q2 (31–38) *B* = 0.1 Q3 (39–46) *B* = −0.4 Q4 (47–59) *B* = 0.4, *P* = 0.30 ^b^	8
Tan et al. (2021) [52]	Non-fasting serum vitamin B_12_ (147–297 pmol/L)	1st and 2nd trimester	No linear association with neonatal HC z-score at birth	=	*B* = −0.00278, *P* = 0.52 ^c^ *B* = −0.00715, *P* = 0.10 ^d^	6
Gadgil et al. (2014) [41]	Plasma vitamin B_12_ (138.6–261.4 pg/mL)	Late pregnancy (mean 36 weeks GA)	No correlation with neonatal HC at birth	=	*R* = 0.22, *P* = 0.28	4
Jiang et al. (2016) [39]	Serum vitamin B_12_ (Not reported)	Not reported	Positively correlated with neonatal HC at birth	+	*R* = 0.511, *P* = 0.029	3
Takimoto et al. (2007) [38]	Non-fasting serum vitamin B_12_ (Not reported)	1st, 2nd, and 3rd trimesters	Association with neonatal HC at birth but effect size too small for physiological significance	=	Not reported	3
Dietary intake						
Neumann et al. (2013) [53]	Dietary vitamin B_12_ intake from quantitative food weighing and dietary recall	From 1st/2nd trimester to term	No associations with neonatal HC at birth	=	Not reported	3
Takimoto et al. (2007) [38]	Daily dietary vitamin B_12_ intake	1st, 2nd, and 3rd trimesters	Not related to neonatal HC at birth	=	*ES* = −0.003, *P* = 0.20 ^c^ *ES* = 0.02, *P* = 0.08 ^d^ *ES* = 0.01, *P* = 0.10 ^e^	3

Abbreviations: effect size (ES), gestational age (GA), head circumference (HC), Q (quartile). Significant associations between the exposure and the outcome are reported as + if positive and − if negative. = indicates no significant association. ^a^ total vitamin B_12_, ^b^ active vitamin B_12_, ^c^ first trimester, ^d^ second trimester, ^e^ third trimester.

**Table 4 genes-12-01634-t004:** Description and summary of 6 studies that investigated associations between maternal tHcy concentrations during pregnancy and newborn head measurements ordered according to the quality score.

Author (Year)	Exposure (Range)	Exposure Timing	Outcome(s)	Association	Statistical Value	Quality Score
Bergen et al. (2016) [40]	Plasma tHcy (4.9–10.5 µmol/L)	Early pregnancy (median 13.5 weeks GA)	≥8.31 µmol/L tHcy associated with reduced foetal HC (−1.6 mm) from late pregnancy (median 30.4 weeks GA) to birth	=	*Β* = −1.6, *P* = 0.06	8
Nilsen et al. (2010) [37]	Plasma tHcy (<4.5->5.8 µmol/L)	Mid-pregnancy (median 18 weeks GA)	No linear association with neonatal HC at birth	=	*P* = 0.48	7
Tan et al. (2021) [52]	Non-fasting serum tHcy (4.5–5.7 µmol/L)	1st and 2nd trimester	No linear association with neonatal HC z-score at birth	=	*B* = 0.0202, *P* = 0.65 ^a^ *B* = 0.0762, *P* = 0.06 ^b^	6
Takimoto et al. (2011) [31]	Non-fasting plasma tHcy (Not reported)	3rd trimester	Positively correlated with neonatal HC at birth	+	*R* = 0.53, *P* < 0.01	4
Jiang et al. (2016) [39]	Serum tHcy (Not reported)	Not reported	Negative correlation with neonatal HC at birth	−	*R* = −0.401, *P* = 0.034	3
Takimoto et al. (2007) [38]	Non-fasting plasma tHcy (Not reported)	1st, 2nd, and 3rd trimesters	No relation with neonatal HC at birth except for the second trimester	= (+ second trimester tHcy)	*ES* = 0.02, *P* = 0.92 ^a^ *ES* = 1.54, *P* = 0.03 ^b^ *ES* = −0.24, *P* = 0.53 ^c^	3

Abbreviations: effect size (ES), head circumference (HC), total homocysteine (tHcy). Significant associations between the exposure and the outcome are reported as + if positive and − if negative. = indicates no significant association. ^a^ first trimester, ^b^ second trimester, ^c^ third trimester.

**Table 5 genes-12-01634-t005:** Description and summary of three studies that investigated associations between maternal dietary patterns during pregnancy, one-carbon metabolism, and foetal or neonatal head measurements ordered according to the quality score.

Author (Year)	Exposure	Exposure Timing	Association with One-Carbon Metabolism	Outcome(s)	Association	Statistical Value	Quality Score
Timmermans et al. (2012) [57]	Mediterranean diet	Early pregnancy (median 13.5 weeks GA)	Low adherence was associated with high tHcy, low serum vitamin B_12,_ and folate	Low adherence associated with a smaller foetal HC in late pregnancy	+	Difference in SDS = −0.08, *P*= 0.01	7
Lecorguillé et al. (2020) [52]	Varied and balanced diet	2nd trimester (average 15 weeks GA)	High positive coefficients for B vitamins, choline, and methionine	Not associated with neonatal HC at birth	=	*B* = 0.01, *P* = 0.43	6
Parisi et al. (2018) [56]	Dairy-rich diet	1st trimester (≤8 + 0 weeks GA)	Associated with lower plasma tHcy and correlated to vitamin B_2_, B_6,_ and B_12_	Associated with increased foetal TCD measurements in 1st and 3rd trimester	+	*B* = 0.02, *P* < 0.01	6

Abbreviations: gestational age (GA), head circumference (HC), total homocysteine (tHcy), transcerebellar diameter (TCD). Significant associations between the exposure and the outcome are reported as + if positive and − if negative. = indicates no significant association.

**Table 6 genes-12-01634-t006:** Reference values for one-carbon metabolism intermediates among the general and pregnant population based on the World Health Organization and the National Institute of Health in The Netherlands.

	Reference Values		References
	General Population	Pregnancy	
Serum/plasma folate (nmol/L)	13.5–45.3	>10 *	[42]
RBC folate (nmol/L)	>340 **	>906	[42,66]
Serum/plasma vitamin B12 (pmol/L)	130–700	>150 ***	[67,68]
Homocysteine (µmol/L)	<15	The lower, the most optimal for pregnancy health	[68]

* <10 nmol/L indicates folate deficiency, ** <340 nmol/L indicates folate deficiency, *** <150 pmol/L indicates vitamin B12 deficiency. Abbreviations: red blood cell (RBC).

## Data Availability

Not applicable.

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
