# Peer review of "Maternal One-Carbon Metabolism during the Periconceptional Period and Human Foetal Brain Growth: A Systematic Review"

_genes, 2021, doi:10.3390/genes12101634_

Round 1
Reviewer 1 Report
The importance of adequate folate levels in the first three weeks of pregnancy as a prevention for NTDs is well recognised. This study examines the broader implications of one-carbon metabolic pathway aberrations and the implications for the child’s brain development and subsequent behavioural and language abilities.
Overall, the review is very detailed which perhaps takes from a general overview which can be preferred when a review is sought. The study selection is rather tedious and a little concerning when only 2% of the papers can be reviewed after such a screen. Are the authors being too stringent?
The manuscript is disjointed partly due to the lengthy tables – could they be put into supplementary?
The summary/figure 3 should be improved with images (drawings) with more details of the conclusions.
Minor:
Line 39 dynamically
Line 45 remove the before one-carbon
Line 56 in instead of to
Line 59 increased total homocysteine
Figure 1 – purine synthesis
Line 354 values of one-carbon
Line 368 ranges
Author Response
Overall, the review is very detailed which perhaps takes from a general overview which can be preferred when a review is sought. The study selection is rather tedious and a little concerning when only 2% of the papers can be reviewed after such a screen. Are the authors being too stringent?
Answer: The literature search strategy was performed using broad terms in order to avoid missing any articles, since the topic is not studied extensively. This resulted in a large amount of articles which had to be excluded as we carefully selected those relevant to our research question. For this reason, it can appear that we were stringent but in reality, we just wanted to avoid missing any articles.
The manuscript is disjointed partly due to the lengthy tables – could they be put into supplementary?
Answer: Table 1 was moved to the supplementary data. The other tables were kept in the text but moved to separate pages (in horizontal view) to avoid the result section from being disjointed.
The summary/figure 3 should be improved with images (drawings) with more details of the conclusions.
Answer: Figure 3 was made again, referring to the results obtained from the review.
All other minor comments were changed too.
The track changes made the document quite unclear, so I uploaded the version without track changes. I can send the version with track changes if needed. The submission button does not allow me to load the supplementary file.

Reviewer 2 Report
The paper by Rubini et al provides a systematic review of the literature concerning the effects of one-carbon metabolism during the periconceptional period and later pregnancy on human foetal and neonatal brain development and growth, with a view to identify possible origins of neurodevelopmental disorders. The study is the first in this precise direction and, in my view, has been conducted with both considerable care and attention to detail in design and execution. Whilst it is difficult to draw many solid conclusions given the scarcity of suitable previous studies and confounders and the varied demographics of maternal subjects included, I think the use of growth criteria to assess outcomes has been wisely selected. The exclusion criteria and quality assessment of selected studies is also appropriate. Overall, their findings will be of significant interest to those with clinical responsibilities in neurodevelopmental disorders. I commend the authors for their thorough and excellent work.
I wondered whether in the Discussion, the authors could briefly address three queries. First, what is the impact of folate excess rather than just deficiency? Supplementary levels above recommended are referred to in the text but can any conclusion be drawn on effect on growth criteria assessed? Also, what impact will maternal population demographics have on outcomes such as maternal age, metabolic status, ethnicity etc? Lastly, can the depth of the study be increased by consideration, where relevant, of combinatorial interactions across micronutrients, for example are folate effects influenced by availability of other measured factors?
Author Response
All comments were processed and added to the discussion (please see attachment). This version also includes the processed comments from the other reviewer.
The track changes made the document quite unclear, so I uploaded the version without track changes. I can send the version with track changes if needed. The submission button does not allow me to load the supplementary file.
